# Dietary Intake of Flavonoids Associated with Sleep Problems: An Analysis of Data from the National Health and Nutrition Examination Survey, 2007–2010

**DOI:** 10.3390/brainsci13060873

**Published:** 2023-05-29

**Authors:** Lingman Wang, Jianxiong Gui, Ran Ding, Xiaoyue Yang, Jiaxin Yang, Hanyu Luo, Dishu Huang, Ziyao Han, Li Jiang

**Affiliations:** Department of Neurology, Children’s Hospital of Chongqing Medical University, National Clinical Research Center for Child Health and Disorders, Ministry of Education Key Laboratory of Child Development and Disorders, Chongqing Key Laboratory of Pediatrics, Chongqing 400014, China

**Keywords:** flavonoid intake, sleep duration, sleep disorders, NHANES

## Abstract

Flavonoids possess the latent ability to protect against sleep disorders. We examined the correlation between daily flavonoid intake and sleep duration, and sleep disorders. We enrolled 8216 participants aged ≥ 20 from the National Health and Nutrition Examination Survey (NHANES, 2007–2010), carrying out a cross-sectional study. Flavonoid intake was collected by dietary intake interview recalls. Logistic regression was utilized to evaluate the association between flavonoid intake sleep duration, and sleep disorders. We used subgroup and interaction analysis to explore differences between subgroups. When adjusting covariates in model 2, anthocyanidins, flavan-3-ols, flavones, flavonols, and the sum of flavonoids were considerably related to insufficient sleep duration (odds ratio (OR) (95% confidence interval (CI)); 0.83 (0.72, 0.95); 0.91 (0.83, 0.98); 0.63 (0.41, 0.98); 0.78 (0.64, 0.94); 0.85 (0.76, 0.95), respectively); the converse association was observed between flavanones, and flavones and the risk of sleep disorders (OR (95% CI); 0.85 (0.77, 0.95); 0.61 (0.41, 0.90), respectively). In relation to insufficient sleep, there were statistically significant interactions between flavonoid consumption and race/ethnicity, and education level. In relation to insufficient sleep, there were statistically significant interactions between flavonoid consumption and working status. In this study, we found that certain flavonoids were linked to increased sleep duration and a lower risk of sleep problems. Our research indicated that flavonoids might be a preventive factor for sleep disorders.

## 1. Introduction

As encompassing one-third of human life [1], sleep is an essential physiological requirement for human function and determines an individual’s overall health [2]. Sleep is an emergent set of many metabolic functions that are primarily neurobiologically regulated and have an impact on many physiologic systems [3]. Numerous people experience sleep problems, such as sleep deprivation and poor sleep quality, which have severe health consequences [4]. Noteworthily, sleep disturbances are closely entwined with a sequence of poor health outcomes. Sleep disorders and insufficiency are frequent predictors of child/adolescent mental diseases, such as anxiety, depression, and even suicidal ideation [5]. Meanwhile, sleep disorders contain insomnia, sleep-related breathing disorders, central disorders of hypersomnolence, circadian rhythm sleep-wake disorder, sleep-related movement disorders, parasomnias, and other-sleep disorders [6]. The diagnosis of insomnia during office visits in America experienced an 11-fold increase, going from 800 thousand to 9.4 million between 1993 and 2015 [7]. Despite the fact that 7 or more hours are recommended for adults to support optimal health [8], 40% of adults in the U.S. sleep fewer 7 seven hours per night [9]. The reduced sleep duration was presumably owing to prolonged working hours, less reliance on daylight hours, and lifestyle changes. Shankar et al. revealed that shorter and longer sleep duration increased the incidence of self-rated health unfavorable outcomes [10]. Chronic issues include heart disease, kidney disease, hypertension, obesity, diabetes, and mental illness co-occur with little sleep [11,12,13,14]. Insufficient sleep increases the risk of developing multiple chronic conditions that are also correlated with the onset of sleep disorders. Urgent exploration is required to determine the protective variables for improving sleep disorders. Recently, phytochemicals have been attributed to alleviating sleep disorders. Compared with traditional pharmacotherapy including tranquilizers and hypnotics, herbal medicines have fewer side effects. The flavonoids extracted from the stems and leaves of peanuts could reduce neuronal excitability to improve sleep quality [15]. There were few studies on the relationship between isoflavone intake and sleep. The inherent beneficial function of soy isoflavone intake in adult sleep regulation was supported by a Chinese longitudinal analysis [16]. A randomized controlled trial found that consuming isoflavones for four months improved sleep quality and decreased the frequency of insomnia in postmenopausal women [17].

Flavonoids serve as bioactive, nutritional, and functional compounds in various plants, having polyphenolic structures [18]. The available sources of flavonoids are fruits, vegetables, tea, dark chocolate, and certain beverages [19,20]. Flavonoids represent C6-C3-C6 rings. Specifically, two benzene rings (ring A and ring B) are connected by three-carbon-ring C [21]. According to the heterocycle (ring C) variations, flavonoids can be categorized into six subclasses: isoflavones, flavonols, flavan-3-ols, flavanones, flavones, and anthocyanins [22]. The literature has recently placed a significant focus on flavonoids, unveiling a multitude of potential valuable effects. Flavonoids provide numerous health advantages for human beings, notably anti-inflammatory, antiviral, anti-cancer, and other biological effects [23,24]. Evidence from the molecular mechanism studies supported that flavonoids suppressed the activities of inflammatory immune cells and inflammatory genes and mediators as well as modulating pro- or anti-inflammatory signaling pathways [25]. Flavonoids, in fact, contain different quantities of phenolic hydroxyl groups that can act as hydrogen donors to reduce free radicals, aid in reducing peroxidation damage to the body, and enhance the levels of endogenous antioxidants [26]. It has been demonstrated that flavonoids were capable of inhibiting the MAPK signaling pathway that controls the expression of nitric oxide synthase (iNOS) and cyclooxygenase (COX)-2 induced by lipopolysaccharide (LPS) [27]. Notably, a class of studies indicated that flavonoids exerted beneficial effects on reducing some diseases, for example, metabolic disorders, cancer, Alzheimer’s disease, and Parkinson’s Disease [28,29,30]. Furthermore, the dietary intake of flavonoids could protect neuronal cells against death in both oxidative stress- [31] and amyloid-β(Aβ)-induced-neuronal death models [32]. However, the association between flavonoid consumption and sleep problems has yet to be adequately investigated.

Thus, we investigated whether dietary flavonoids were related to the risk of sleep problems in individuals recruited in the National Health and Nutrition Examination Survey (NHANES) from 2007 to 2010. Our hypothesis was that the consumption of flavonoid-rich foods at a population level is linked with prevention of sleep problems.

## 2. Materials and Methods

### 2.1. Study Population

National Health and Nutritional Examination Surveys (NHANES) is a cross-sectional survey to investigate the health and nutritional status of the U.S. population conducted by the National Center for Health Statistics (NCHS). The survey adopts a stratified multi-stage sampling design; data are available from the interviews and physical examinations of participants. A total of 12,153 participants older than 20 years from the NHANES 2007 to 2010 participants were engaged in this study. The survey utilizes a multiple-stage, stratified sampling of probability procedure, with counties, blocks, homes, and individuals within households being chosen at random. Next, we excluded enrollments missing information on social demography, flavonoid intake, and evaluation of sleep disorders, BMI, alcohol consumption state, and work. Consequently, we included 8189 adults aged more than 20 years (Figure 1). All participants in the NHANES gave informed consent.

### 2.2. The Intake of Flavonoids

The consumption of individual flavonoids was estimated by 2-days dietary recall details from the dietary intake interview component of the NHANES 2007–2010. The 2-days dietary recall refers to the subject’s food intake being documented throughout an entire day during the self-reported interview, followed by a telephone conversation to get information on the participant’s food intake on the second day. The average consumption from both occasions is then utilized to assess individual flavonoid compound intake. To record dietary intakes and determine nutrients for the NHANES, the United States Department of Agriculture (USDA) uses the Food and Nutrient Database for Dietary Studies (FNDDS). Using data from the Flavonoid Database from FNDDS versions 4.1 and 5.0, analytical values for 29 flavonoids as well as class totals were provided. Flavonoids were categorized into six classes: isoflavones, anthocyanidins, flavan-3-ols, flavanones, flavones, and flavonols. Six classes of flavonoids make up the total amount of flavonoids.

### 2.3. Sleep Problems

Sleep disorders and sleep duration were obtained through interviews and self-reporting. Sleep duration was obtained through questionnaire interviews (e.g., how much sleep do you usually get at night on weekdays or workdays?) Two categories of sleep were distinguished: normal (7–8 h per day), insufficient (<7 h per day), or excessive (>8 h per day). The sleep disorders configuration was questioned for the descriptive questions SLQ050 and SLQ060: “Have you previously been informed by a physician or other health practitioner that you had trouble sleeping?” and “Have you previously been informed by a physician or other health practitioner that you have a sleep disorder?”. Those who answered “yes” to any one of two questions were later regarded to have sleep disorders in further analysis.

### 2.4. Covariates

Age, sex (male, female), race/ethnicity (non-Hispanic White, non-Hispanic Black, Mexican-American, others), educational level (less than high school, completed high school, or more than high school), alcohol consumption (no, mild, moderate, and heavy), smoking status (never smokers, former smokers, and current smokers), working status (not working, looking for work, regular daytime schedule, regular evening or night shift, rotating shift, or other) were collected via a structured questionnaire. Poverty status was judged by the income-to-poverty ratio of <1 (below the poverty threshold) versus ≥1 (reference). The participant’s height and weight were used to calculate BMI (kg/m^2^) during the physical examination. The participants were classified according to the categorization suggested by the World Health Organization [33], which includes overweight (BMI 25 to <30 kg/m^2^) and obesity (class I: 30 to <35 kg/m^2^; class II: 35 to <40 kg/m^2^; and class III: ≥40 kg/m^2^).

### 2.5. Statistical Analysis

Due to the complexity of the multiple-stage surveys, sample weights were applied for all statistical analyses according to the R version 4.2.1, with *p* < 0.05 on both sides considered statistically significant.

The means (standard error) or numbers (weighted percentage) were employed to describe the data. We conducted the logistic regression analyses to evaluate both the link between flavonoid intake and sleep time and the link between flavonoid intake and sleep disorders, as well as calculating prevalence odds ratios (ORs) and 95% confidence intervals (CIs). The multinomial logistic regression models were used to assess the relationship between flavonoids and sleep duration, as well as sleep disorders. Each component of flavonoids was the only independent variable in model 1, and model 2 was adjusted for age, sex, race, poverty status, educational level, smoking status, alcohol consumption, BMI, and working status.

Next, we used the subgroup and interaction analyses to examine both the correlation between insufficient sleep and a total amount of flavonoids and the connection between sleep disorders and a total amount of flavonoids varied among subgroups.

## 3. Results

Table 1 lists the demographic characteristics and flavonoid consumption of the participants. There were 8216 participants in the study, including 3978 men and 4238 women. The average age of participants was 46.82 years old, and non-Hispanic White was the predominant race/ethnicity (51.56%). The average sleep duration of the participants was 6.89 h, and the incidence of sleep disorders was 26.46%.

As shown in Figure 2, the intake of flavonoid levels had inverse relationships with insufficient sleep (≤6 h/night) in model 1. After further adjusting covariates in model 2, anthocyanidins, flavan-3-ols, flavones, flavonols, and the sum of flavonoids were considerably related to insufficient sleep duration (OR (95% CI); 0.83 (0.72, 0.95); 0.91 (0.83, 0.98); 0.63 (0.41, 0.98); 0.78 (0.64, 0.93); 0.85 (0.76, 0.95), respectively). Conversely, there seem to be no correlations between flavonoid dietary intake and excessive sleep (≥9 h/night) in model 1, except for flavanones. In model 2, all classes of flavonoid intake were not linked with excessive sleep (Figure 2).

As shown in Figure 3, the multivariable logistic regression analyses displayed that flavanones and flavones were substantially reverse associated with the likelihood of sleep disorders (OR (95% CI); 0.85 (0.77, 0.94); 0.60 (0.44, 0.82), respectively). When adjusting covariates in model 2, the converse association was observed between flavanones, and flavones and the risk of sleep disorders (OR (95% CI); 0.85 (0.77, 0.95); 0.61 (0.41, 0.90), respectively). However, isoflavone intake was observed to be positively correlated with sleep disorders (OR (95% CI); 1.29 (1.04, 1.59).

Figure 4 shows subgroup and interaction analysis between insufficient sleep and the sum of flavonoids. There were statistically significant interactions between flavonoid consumption and race/ethnicity, and educational level in connection to insufficient sleep (*p* = 0.01, *p* = 0.002, respectively). There was a statistical link between insufficient sleep and dietary intake of total flavonoids among the subgroups “20–40 years”, “male”, “female”, “non-Hispanic White”, “poverty income ratio ≥1”, “completed high school”, “moderate alcohol consumption”, and “not working”.

Figure 5 shows subgroup and interaction analysis between sleep disorders and a sum of flavonoids. There were statistically significant interactions between flavonoid consumption and working status in connection to sleep disorders (*p* = 0.03). Beyond that, there was a statistical link between sleep disorders and dietary intake of total flavonoids among the subgroups “completed high school” and “looking for work”.

## 4. Discussion

To date, our study was the first assessment of the link between flavonoid intakes and incidence of insufficient sleep and the risk of sleep disorders among adults employing a sizable, nationally representative sample from the NHANES 2007–2010.

Sleep problems tend to be the most prevalent illnesses, and they are frequently disregarded. The consequences of sleep loss and sleep disorders extend far beyond the individual, with widespread implications for public health. These consequences include higher mortality and morbidity rates, greater risk of accidents and injuries, lower family well-being, inferior functioning and quality of life, and increased healthcare utilization [34]. There is a pressing need for the investigation of factors that can be modified to lower the likelihood of sleep-related problems. Flavonoids are plentiful in diverse medicinal plants and are widely distributed in the food system [35]. Flavonoids’ exceptional biological activities and nutritional properties sparked our interest in exploring their pharmacological effects on sleep. It has been demonstrated that flavonoid therapy could improve sleep problems by having fewer negative effects [15], which makes us wonder whether flavonoid intakes are also connected with sleep duration and sleep disorders.

The current analysis identified that the occurrence of insufficient sleep was linked to fewer anthocyanidins, flavan-3-ols, flavones, flavonols, and the sum of flavonoids when adjusted for certain potential covariates. The study also revealed that flavanones and flavones were statistically inversely linked to the risk of sleep disorders. In animal experiments, flavonoids have been discovered to have sedative and hypnotic effects, such as reducing sleep latency and improving sleep quality [36,37]. Besides, procyanidin B2 from lotus seedpod was inferred to treat insomnia in rats [38]. A flavonoid compound from Mulberry had favorable effects on sleep perturbation [39]. Chamomile dried flowers contained many flavonoids and were studied for preliminary efficacy and safety in improving sleep and daytime symptoms in patients with chronic insomnia by Zick et al. [40]. They concluded that chamomile treatment might have modest advantages for daytime performance and mixed benefits on sleep diary measurements. Some studies have identified the relationships between isoflavone and sleep. It has been documented that isoflavone treatment could alleviate the symptoms of postmenopausal women with insomnia [17]. Our study, however, did not find similar results between isoflavone intake and insufficient sleep and sleep problems. It could be because of differences in the research population’s baseline characteristics. The population Hachul et al. studied was a total for postmenopausal women aged 50–65 years old. Contrarily, both adult males and females were included in our study. Apart from that, it could also result from varied symptoms of sleep problems while insomnia is one of the manifestations. In addition to isoflavones, we studied the link between flavonoid intake and its improving potential on sleep disruption and disorders among adults aged more than 20 for the first time, which revealed the health-promoting function of flavonoids.

Interestingly, our subgroups and interaction analysis revealed that non-Hispanic White high school graduates had a stronger correlation between the total amount of flavonoids and insufficient sleep. A more pronounced correlation between flavonoids and inadequate sleep duration has been observed in populations with a poverty index exceeding 1. Moreover, our results indicated a stronger link between overall flavonoid consumption and sleep disorders among those who were looking for work. The effects of ethnic polymorphism, economic patterns, and varying stress levels on the relationship between flavonoids and sleep quality may provide an explanation for these outcomes. Research suggested that employment, education, and income exhibited a negative correlation with insomnia disorder [41,42]. Non-Hispanic Whites may be more sensitive to the effects of flavonoids; however, further research is required to substantiate this. Those with higher levels of education may work longer hours at stressful jobs, which are additional factors that may impact sleep. In comparison, those with lower levels of education may experience more psychological discomfort [43]. Diverse daily dietary patterns among individuals of different ethnicities and varying poverty levels may lead to differences in flavonoid consumption that can modify the association between flavonoids and inadequate sleep or sleep disorders. People who were searching for employment could suffer anxiety and stress, which could lead to poor sleep quality.

Nevertheless, the underlying mechanisms that drive the connection between flavonoid intake and sleep are unidentified; some possible explanations exist. Some studies noted the possibility that the effect of flavonoids on sleep might be mediated by inflammation and oxidative stress. An increasing amount of evidence implied that sleep deprivation was linked with the elevation of inflammatory proteins such as IL-6, IL-10, TNF-α, and CRP [44,45]. As sleep deprivation or loss may cause oxidative stress, it has been suggested that sleep’s anti-oxidative role was an adaptive reaction [46]. Trials from different animal models also indicated that sleep deprivation stimulated oxidative stress [47]. Furthermore, growing evidence showed that flavonoids could regulate the effects of anti-inflammatory and anti-oxidative stress activity. A flavone molecule called luteolin was said to have a substantial anti-inflammatory effect as its primary pharmacological mechanism [48]. Two flavonoids from citrus species, hesperidin (Hsd) and hesperetin (Hst), are beneficial for lowering oxidative stress and inflammation [49]. Moreover, short sleep duration and sleep disorders are linked to an imbalance in the intestinal microbiome [50]. It has been recommended that natural flavonoid intake, such as anthocyanins, could perhaps regulate gut dyshomeostasis [51]. Shrestha et al. reported that flavonoid components might have a strong affinity with gamma-aminobutyric acid (GABAA)/benzodiazepine receptors, resulting in decreased sleep latency and increased sleep quality in mice [52]. Since it has been testified that members of flavonoids can traverse the blood–brain barrier (BBB) and arrive at the central nervous system [53,54], flavonoids own natural neuroprotection to modulate sleep.

The merits of our study included the relatively large sample size and the high response rate for an NHANES national representative survey. Indeed, FNDDS has currently updated data on flavonoid intake for the 2017–2018 cycle. Unfortunately, there are differences in the sleep-related questionnaires between NHANES 2017–2018 cycle and the ones from previous years of the NHANES year cycles (Appendix A), as well as NHANES 2017–2018 cycle, which provided only about 2800 participants who had 2-days dietary recall information available to estimate flavonoid intake. Hence, we chose the participants from NHANES 2007–2010 cycle as our study population to meet the accuracy and timeliness of data. To our knowledge, we probed the link between flavonoid intake and sleep duration and sleep disorders in adults over 20 years in the United States for the first time. We performed logistic analyses, controlled for potential confounding factors, and identified the connections. To estimate the consumption of individual flavonoids at an individual level, we implement the 2-days dietary recall method. Compared to a 1-day recall, the 2-days dietary recall to some extent reduces the possibility of chance variation and better reflects an individual’s dietary intake. This study has some limitations that also need to be taken into account as well. First of all, because cause-and-effect conclusions could not be drawn from the NHANES cross-sectional investigations, we were unable to ascertain whether elevated flavonoid intake enhanced sleep duration or whether sleep duration improved flavonoid intake. So, cohort studies are necessitated to confirm the effect. Second, self-reported doctor diagnoses were used for the appraisal of sleep disorders, which could bring recall bias as a consequence. Similarly, self-reported sleep duration may need to be more accurate, whereas operating an objective sleep measurement method such as polysomnography in large-scale experiments is challenging. Third, the 2-days dietary recall has restrictions since a portion of participants dropped out before the second dietary interview and did not provide data for the second interview. Fourth, some individual dietary habits of flavonoid consumption could influence the results of our study. Finally, we cannot conclude whether a particular type of sleep problem was related to flavonoid intake due to the challenges in data collecting and matching.

## 5. Conclusions

In summary, our study investigated the link between flavonoid intake and the incidence of insufficient sleep, and the risk of sleep problems among adults aged over 20. Lack of sleep is linked with lower levels of anthocyanidins, flavan-3-ols, flavones, flavonols, and the total amount of flavonoids. Statistics showed a negative correlation between the risk of sleep problems and flavanones and flavones. Notably, the link between inadequate sleep and the sum of flavonoids was more significant among non-Hispanic White high school graduates. The link between sleep disorders and the sum of flavonoids was more significant among those who were looking for work. These findings could be explained by how different degrees of stress, ethnic polymorphism, and economic patterns affect the quality of sleep. These findings could pave the way for future intervention studies into potential treatments or preventative measures for sleep disorders, such as the effects and doses of specific flavonoids.

## Figures and Tables

**Figure 1 brainsci-13-00873-f001:**
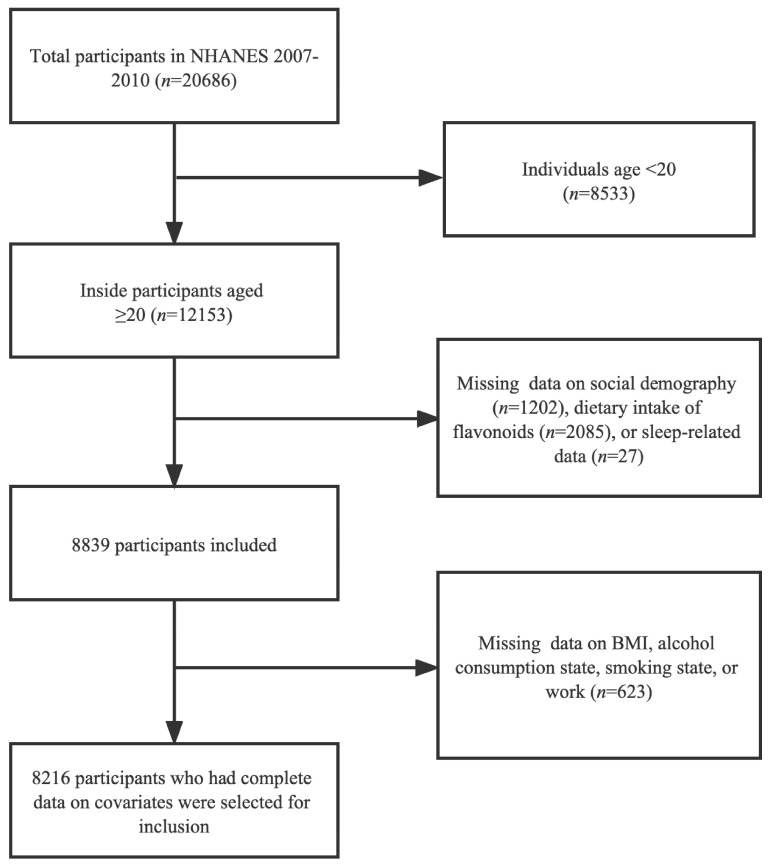
Flow diagram of the inclusion of participants.

**Figure 2 brainsci-13-00873-f002:**
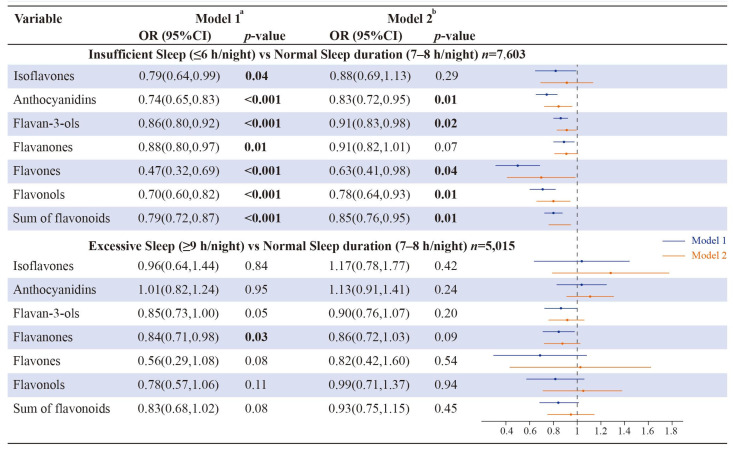
Association of insufficient sleep (*n* = 7603) and excessive sleep (*n* = 5015) with flavonoid intake in NHANES 2007–2010 participants. The numbers represented in bold font indicate statistical significance at a level of *p* < 0.05 for both tails. Model 1 ^a^ was a crude model with no adjusted covariates; Model 2 ^b^ was adjusted for sex, age, race/ethnicity, poverty status, education, smoking status, alcohol consumption, and BMI.

**Figure 3 brainsci-13-00873-f003:**
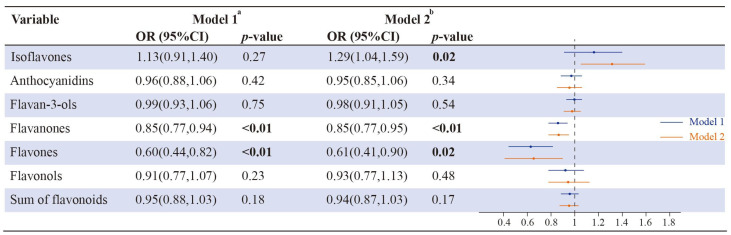
Adjusted multinomial logistic regression (OR and 95% CI) of sleep disorders with flavonoid intake in NHANES 2007–2010 participants (*n* = 8216). The numbers represented in bold font indicate statistical significance at a level of *p* < 0.05 for both tails. Model 1 ^a^ was a crude model with no adjusted covariates; Model 2 ^b^ was adjusted for sex, age, race/ethnicity, poverty status, education, smoking status, alcohol consumption, and BMI.

**Figure 4 brainsci-13-00873-f004:**
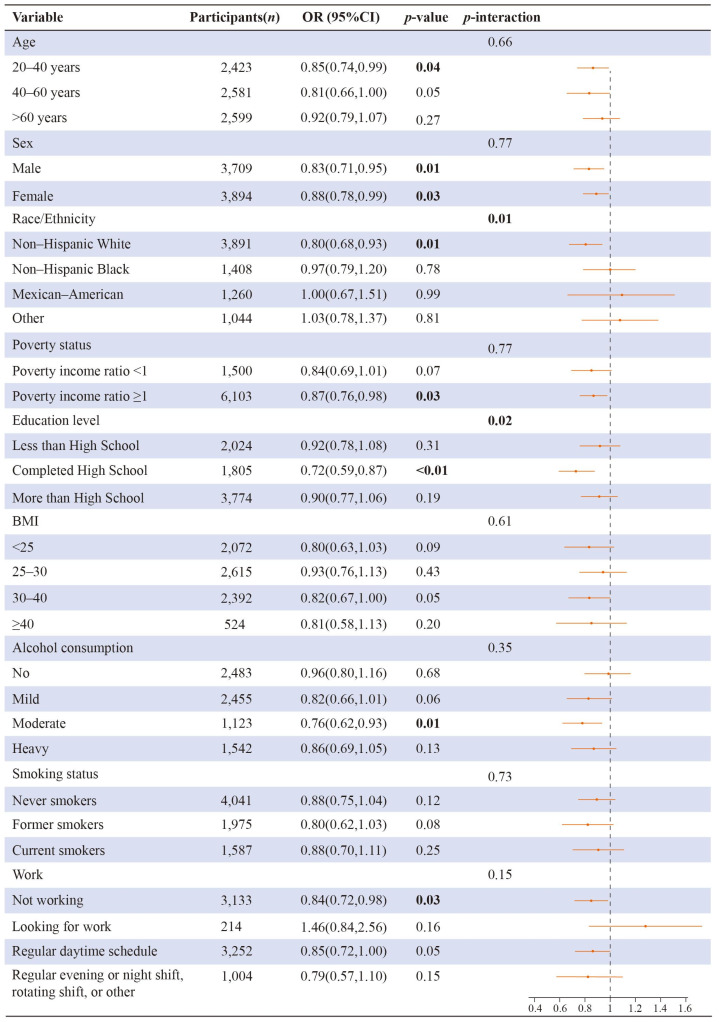
Subgroup and interaction analysis between insufficient sleep and a sum of flavonoids. The numbers represented in bold font indicate statistical significance at a level of *p* < 0.05 for both tails. The model was adjusted for the potential covariates except the grouping variable.

**Figure 5 brainsci-13-00873-f005:**
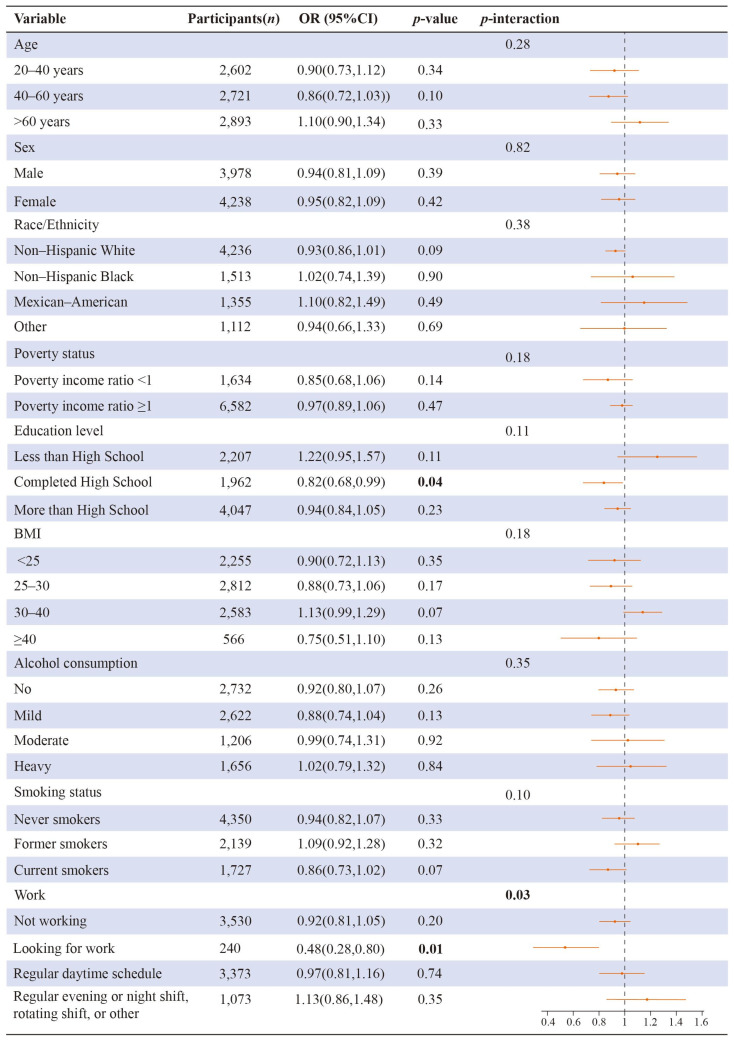
Subgroup and interaction analysis between sleep disorders and a sum of flavonoids. The numbers represented in bold font indicate statistical significance at a level of *p* < 0.05 for both tails. The model was adjusted for the potential covariates except the grouping variable.

**Table 1 brainsci-13-00873-t001:** Characteristics of subjects (age ≥ 20 years) from NHANES 2007–2010 (*n* = 8216).

Variable	Mean (SE) or *n* (Weighted Percentage)
Age (years)	46.82 (0.38)
BMI (kg/m^2^)	28.87 (0.12)
Sleep duration (hours)	6.89 (0.03)
Isoflavones (mg)	1.71 (0.15)
Anthocyanidins (mg)	14.30 (0.92)
Flavan-3-ols	169.72 (9.39)
Flavanones (mg)	13.38 (0.60)
Flavones (mg)	0.99 (0.06)
Flavonols (mg)	18.12 (0.52)
Sum of flavonoids (mg)	218.22 (9.97)
Sex	
Female	4238 (51.58)
Male	3978 (48.42)
Race/Ethnicity	
Non-Hispanic White	4236 (51.56)
Non-Hispanic Black	1513 (18.42)
Mexican-American	1355 (16.49)
Other	1112 (13.53)
Poverty status	
Poverty income ratio <1	1634 (19.89)
Poverty income ratio ≥1	6582 (80.11)
Education level	
Less than High School	2207 (26.86)
Completed High School	1962 (23.88)
More than High School	4047 (49.26)
Alcohol consumption	
No	2732 (33.25)
Mild	2622 (31.91)
Moderate	1206 (14.68)
Heavy	1656 (20.16)
Smoking status	
Never smokers	4350 (52.95)
Former smokers	2139 (26.03)
Current smokers	1727 (21.02)
Work	
Not working	3530 (42.96)
Looking for work	240 (2.92)
Regular daytime schedule	3226 (39.26)
Regular evening or night shift, rotating shift, or other	1220 (14.85)
Sleep disorders	
Yes	2174 (26.46)
No	6042 (73.54)

Continuous data were displayed as weighted means (standard errors), whereas categorical variables were exhibited as unweighted numbers (weighted percentages). NHANES, the national health and nutrition examination survey; BMI, body mass index.

## Data Availability

Publicly available datasets were analyzed in this study. This data can be found here: [https://www.cdc.gov/nchs/nhanes/index.htm] (accessed on 20 March 2023).

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
