# Peer review of "Dietary Intake of Flavonoids Associated with Sleep Problems: An Analysis of Data from the National Health and Nutrition Examination Survey, 2007–2010"

_brainsci, 2023, doi:10.3390/brainsci13060873_

Round 1
Reviewer 1 Report
The introduction part needs to be revised: I did not check all the cited references but I exemplify the following:
In lines 63 – 65, the phrase: “Flavonoids possess several benefits for human health, including anti-oxidative, anti-inflammatory, antiviral, anti-cancer, and other biological activities” was just copied from reference listed as number 22. Proper references are: M.K. Chahar, N. Sharma, M.P. Dobhal, Y.C. Joshi. Flavonoids: a versatile source of anticancer drugs. Pharmacognosy Rev., 5 (2011), pp. 1-12, and W. Ren, Z. Qiao, H. Wang, L. Zhu, L. Zhang Flavonoids: promising anticancer agents Med. Res. Rev., 23 (2003), pp. 519-534, cited in the reference listed as number 22.
In line 74 the phrase: “Ab-induced-neuronal death models” needs to be further explained, it refers to amyloid beta induced neuronal death models, and the reference is Luo, Y, Smith, JV, Paramasivam, V, Burdick, A, Curry, KJ, Buford, JP, Khan, I, Netzer, WJ, Xu, H & Butko, P (2002) Inhibition of amyloid-beta aggregation and caspase-3 activation by the Ginkgo biloba extract EGb761. Proc Natl Acad Sci U S A 99, 12197–12202.
In lines and 134 and 135 the word "connection" is not properly used.It might be preferable to use "relationship" or "correlation".
In Table 1, "Not working or looking for work" is not a good indicator for grouping because the stress level would be different and would cause different sleep disorders.
From line 271 to 274 it seems that the authors did not delete the journal instructions.
Author Response
Big thanks for your kindly suggestions and sufficient patience! Please see the attachment.

Reviewer 2 Report
The article is particularly interesting, especially since flavonoids are recognized for their numerous health benefits and especially for their antioxidant effect. The article is well done, but I would have a suggestion related to the approach to the study, especially since the large number of participants in the study offers the possibility of quite conclusive results. I would have liked the researchers to approach the BMI categories in much more detail. It would have been very interesting to observe the influence on participants with different degrees of obesity and especially on morbid obesity. There are significant differences between overweight and obesity (BMI over 30) and especially BMI over 40.
Author Response

(The authors gave the same response as above.)

Reviewer 3 Report
This paper relates data analyses from the NHANES 2007-2010 survey regarding link between flavonoid intake and sleep quality. First comment is on the choice of the 2007-2010 datasets. These are over 10 years old and more recent surveys have collected similar variables. Data may be obsolete.
Minor comments:
- line 63: "flavones" is a repeat and shold be removed
- line 68: "X.O." is the only abreviation in the document with "dots"
- line 74: write full name for "Ab"
- line 85: use of "sample" is confusing and needs to be clarified
- line 195: "4. Discussion" should be removed from table title
- line 213: use instead "sleep disturbance" or "sleep perturbation"
- line 236: "mounting tests" is unclear, incorrect probably too, better use "increasing amount of evidence"
- line 240: use "indicate" rather than "declared"
- lines 271-274: is it an editorial statement on the content of the discussion section? if so, it should be deleted. if not, the nature of the message is unclear
- line 299: first author of reference 1 in incorrect, please check and correct
Other comments:
- Tables 2 to 5 should indicate size of population in the various subgroups analysed. It would have been interesting as well to include Forest plots to complete the tables. they also lack information on what response is favored below or above 1 for odds ratio
- Discussion: be more precise on what baseline characteristics of the research population differed from this study to that of Hachul et al. (line 221)
- Conclusion: could there be some ethnic polymorphism as well as economical patterns to explain the different patterns of flavonoid x sleep interactions
Author Response

(The authors gave the same response as above.)

Reviewer 4 Report
The authors presented novel and interesting topic. Considering sleep deprivation as frequent issue related to some important health concerns, it is important to have studies like this. Still, I do think minor improvements should be made prior to the publication of this paper.
Although authors at the end of their manuscript list the limitations of their study, the method for dietary intake assessment is not mentioned. In my opinion, authors should explain why particularly 2 day recall was applied and what are the limitations of this method.
Secondly, the discussion section should be more elaborative and explanatory.
Finally, in the statistics, standard deviation should be considered and applied instead of standard error.
Author Response

(The authors gave the same response as above.)

Round 2
Reviewer 3 Report
Authors have given argumented responses to all comments and suggestions and the manuscript now reflects this.
I think that the argument for choosing the 2007-2010 dataset over the 2017-2018 one is solid and it would help to include the statement into the manuscript so that the reader understands that choice.
Author Response
Thank you very much for your kindly suggestions and sufficient patience! Please see the attachment.
